# The Influence of Elastic Support of Component Glass Panes on Deflection and Stress in Insulating Glass Units—Analytical Model

**DOI:** 10.3390/ma17184669

**Published:** 2024-09-23

**Authors:** Zbigniew Respondek

**Affiliations:** Faculty of Civil Engineering, Czestochowa University of Technology, Dąbrowskiego 69, 42-201 Częstochowa, Poland; zbigniew.respondek@pcz.pl

**Keywords:** glass in construction, insulating glass units, elastically supported plates, deflection and stress in IGUs, climatic loads

## Abstract

Insulating glass units (IGUs) are the most common filling for external building envelopes. These elements have many advantages related to the thermal protection of buildings. However, some climatic loads are generated or modified due to the sealed gas cavity between the glass panes. The gas enclosed in the cavities changes its parameters under external load, which affects the operational deflection and stress in an IGU. In most computational models describing this phenomenon, the component panes are assumed to be simply supported on the edge spacer, which is considered a sufficient approximation. This article, which continues previous work, assumes that the component glass panes can be supported elastically at the edges. The parameter describing this connection is rotational stiffness. Based on the theory of linear–elastic plates, coefficients were determined to calculate the change in cavity volume, deflection, and stress in glass panes. Then, the results of calculations of the influence of rotational stiffness and static values in exemplary IGUs of various structures, loaded with changes in atmospheric pressure and wind, are presented. It was found that a feedback loop occurs here. The deflection and stress in elastically supported single panes are lower than in the case of those simply supported. However, the lower susceptibility to deflection of the component panes weakens the gas interaction in the cavity, and the resultant load on these panes increases. The influence of rotational stiffness on the resulting static values may therefore vary. In the analyzed examples, this influence was primarily negative for symmetrical loads and clearly positive for wind loads.

## 1. Introduction

Insulating glass units (IGUs) are a commonly used filling for windows, glass façades, and other external transparent building envelopes. Their characteristic feature is sealed gas-filled cavities located between the glass component panes of the IGU. The idea behind this structure was to limit heat loss through transparent envelopes. In this context, they are constantly being improved, especially multi-glazed systems [1,2,3] or facades with closed cavities [4]. The tightness of the cavities is achieved by using edge spacers [5], most often with double sealing (Figure 1).

However, it turns out that sealing the cavities in IGUs has a certain side effect. Under the influence of climatic factors, the gas entrapped in the gap changes its parameters, which generates loads and deflections of the component glass panes. For example, an increase in atmospheric pressure or a decrease in gas temperature results in a concave form of deflection of the IGU; the reverse action results in a convex form. This is visually observable through the image distortion reflected from glazing, for example, glass facades [6].

The problem of static values in IGUs (both double- and multi-glazed) is well known. The literature describes relevant analytical models. The most famous and often cited articles are those by Feldmeier [7,8], who assumed that the gas in cavities changes its parameters (pressure, temperature, volume) in accordance with the ideal gas law, while the deflection and stress in glass panes can be estimated based on Kirchhoff’s elastic plate theory. Similar assumptions were made in many other analytical and numerical models [6,9,10,11,12,13]. Another approach to using plate theory, based on Betti’s Analytical Method (BAM) and Green’s functions, was used by Galuppi and Royer–Carfagni [14,15,16]. In the engineering approach, the calculation of deflection and stress in IGUs loaded with climatic factors can be estimated according to the European standard [17]. A comparison of selected analytical methods and a proposal to improve calculation efficiency are presented in [18].

Experimental research in this area is difficult and expensive; static quantities in IGUs depend on many factors, often uncertain and changing over time. Nevertheless, the literature describes such studies, most of which have been conducted recently. Tests of the deflection of IGU components in operational conditions are described in [19] and in simulated pressure or temperature changes in [10,20,21]. In [22], Galuppi et al. experimentally verified the BAM model [14] for double-glazed IGUs, in the range of line-distributed and concentrated loads and also at high temperature. Studies of external load sharing in double- and triple-glazed IGUs are described in [23,24], and tests and numerical analyses on this topic, using IGUs with laminated glass, are presented in [25,26]. In [27], this problem was analyzed numerically, taking into account the effect of large deflections, and in [28] for different shapes of IGUs with varying thicknesses of the component glass. Bedon and Amadio [29,30] developed numerical models of sample IGUs, taking into account all seal elements, using previously experimentally tested parameters of these elements.

The computational practice, confirmed by the experimental studies cited above, reveals some general trends regarding the deflection and stress values in IGUs. In the case of loads acting symmetrically, for example, a change in external atmospheric pressure, interaction of the gas confined in the cavities occurs. Due to the change in the volume of this cavity caused by the deflection of the component glass panes, the gas pressure changes, which partially compensates for the external load. This compensation is greater when the component panes are more susceptible to deflection. As a consequence, the resultant load of the component panes, i.e., the pressure difference between the atmospheric air and the cavity, decreases when the linear dimensions (width, length) of the IGUs increase. It has also been shown [7,8] that for each IGU structure, there is a critical width at which the stress in the component panes is maximum. On the other hand, the use of thicker component panes, as well as increasing the distance between the panes (or increasing the total thickness of the cavities), increases the resultant load [31]. A significant increase in load was also observed when IGUs were used with curved glass [32]. In Cwyl et al. [33], reported a case where a composite spacer was “pulled” into the cavity of a bent IGU, probably due to high underpressure generated under winter conditions.

The situation is different in the case of uniformly distributed loads (e.g., wind action). There is a favorable gas coupling of the IGU component panes. The external load is distributed over all the panes, with large IGU dimensions approximately proportional to the stiffness of the component panes [34].

In most analytical and numerical models, as well as in the standard [17], as a sufficiently accurate approximation, the simple support of the component panes at the edges is assumed, i.e., zero support moments at the connection of the component panes with the edge spacer. It is believed that such an assumption is correct from the point of view of structural safety [14], i.e., taking into account the possibility that elastic support leads to a reduction in the maximum stresses in the IGU, which was shown in experimental studies by Buddenberg et al. [20]. The possibility of elastic support for the component panes was also demonstrated in a study by Stratiy [10]. The IGU model embedded in a window frame and loaded with a pressure difference was tested here. The tests indicated that the deflection of the component slab has intermediate values between those estimated for a simply supported and restrained pane. Similar tests (an IGU was placed in a structure simulating a window frame) described in [21] showed that the measured deflection and stress in the center of the panes differed from the theoretical values for simple support, which may suggest elastic support. Regarding the theoretical analysis of the problem, recently, Galuppi and Zacchei [35] published a theoretical and numerical analysis on the transfer of external load between the panes of double-glazed IGUs, assuming their support with elastic beams, depending on the coefficients describing the bending and torsional stiffness of the support beams.

However, the studies described in the literature do not cover all cases of IGU design and possible variants of climatic loads. It is unquestionable that for elastic support, the deflection and stress in the center of a pane are greater for simple support than for elastic support, but provided that the load on this pane remains constant. Meanwhile, the IGU behaves in a specific way—it is expected that reducing the susceptibility to deflection of the component panes with elastic support increases the resultant load. To estimate possible effects, it is necessary to check the behavior of IGUs with different structures and dimensions. Due to the above, it was noticed that there was a need to supplement the author’s computational model [6,11] with the case of elastic support. The previous article [36] proposed an analytical model enabling the estimation of the resultant load and deflection for IGUs with elastically supported component panes, assuming any rotational stiffness. Therefore, this work continues the analysis presented in [36].

Accordingly, this article aims to complement the computational model by estimating stress in IGUs with elastically supported component panes and then to analyze the influence of rotational stiffness on static values in IGUs with various geometric parameters.

The “Materials and Methods” section presents the proposed analytical model for estimating static quantities in IGUs and its numerical validation. The “Results” section contains an analysis of the influence of rotational stiffness on static values in IGUs of various designs. The “Discussion” section explains some issues regarding the practical application of the proposed model and analyzes what factors may influence the rotational stiffness in real IGUs. The article ends with “Conclusions”.

## 2. Materials and Methods

### 2.1. Base Model

As mentioned, the author’s base model for estimating static quantities in climate-loaded IGUs was described in [6,11]. The model assumes conformity of changes in gas parameters in the cavity with the ideal gas law, a linear relationship between the deflection and stress of the component panes, and the simple support of these panes at the edges. The model allows the estimation of static quantities for IGUs with any number of cavities, loaded with pressure and temperature changes and uniform loads per area. The basis of the calculations is to determine the operational gas pressure in individual cavities *p*_op_ [kPa] and then, based on the load balance, the resultant load *q* [kPa] for each component pane. The problem was reduced to solving an appropriate quadratic equation (double-glazed IGU) or a system of quadratic equations (multi-glazed IGU). In the simplest case (Figure 2) of a double-glass IGU loaded with pressure changes, temperature changes, and wind, the operating pressure in the cavity is
(1)$$p_{\mathrm{op}}=\frac{A}{2\cdot B}+\sqrt{{\left(\frac{A}{2\cdot B}\right)}^{2}+\frac{p_{0}\cdot v_{0}\cdot T_{\mathrm{op}}}{B\cdot T_{0}}}$$
with
(2)$$A=(p_{a}+q_{z,ex})\cdot \alpha_{v,\mathrm{ex}}+p_{a}\cdot \alpha_{v,\mathrm{in}}-v_{0}$$
(3)$$B=\alpha_{v,\mathrm{ex}}+\alpha_{v,\mathrm{in}}$$
where
*p*_0_, *T*_0_, *v*_0_—initial parameters of gas in the cavity: pressure [kPa], temperature [K] and volume [m^3^]; parameters of the production process; it is assumed that under these conditions, the resultant load equals 0;*p*_op_, *T*_op_—operating parameters: pressure [kPa] and temperature [K] in the cavity;*p*_a_—current atmospheric pressure [kPa];*α*_v,ex_, *α*_v,in_—proportionality coefficients describing the change in the volume of the cavity caused by the deflection of a given glass under a unit load per area [m^5^/kN];*q*_z,ex_—wind load [kN/m^2^].

In the above formulas, the index “ex” denotes the pane on the outside air side, and the index “in” denotes the pane on the room side. If necessary, a sign convention is used—load and deflection values are considered to be positive if they face the interior (Figure 2).

The proportionality coefficients *α*_v_ were determined on the basis of the deflection function of a simply supported plate [37] according to the following formula:(4)$$\alpha_{v}={\alpha^{\prime }}_{v}\cdot \frac{a^{6}}{D}\;\;\;with\;\;\;D=\frac{E\cdot d^{3}}{12\cdot \left(1-\mu^{2}\right)}$$
where
*α*′_v_—dimensionless coefficient [−];*a*—width of the glass pane [m];*D*—flexural rigidity of glass pane [kNm];*E*—Young’s modulus of glass [kPa];*d*—glass pane thickness [m];*μ*—Poisson’s ratio of glass [−].

After calculating the resultant load *q* [kPa], the deflection in the center of the glass pane *w*_max_ [mm] can be calculated using the following formula:(5)$$w_{max}={\alpha^{\prime }}_{w}\cdot \frac{q\cdot a^{4}}{D}\cdot 1000$$
where
*α*′_w_—dimensionless coefficient [-].

The article [11] also provides dimensionless coefficients for calculating stress in glass panes.

### 2.2. Deflection and Change of Cavity Volume in IGUs with Elastically Supported Glass Panes

Previous work [36] described an analytical model, taking into account the elastic support of component panes around the perimeter for estimating the resultant load *q* and maximum deflection *w*_max_ in climatically loaded IGUs. The problem was reduced to determining dimensionless coefficients *α*′_v_ and *α*′_w_ for any rotational stiffness of the connection of the glass pane with the spacer (support beam). Further calculations are carried out in accordance with the base model. The analysis of an elastically supported plate is modeled on the rigidly clamped plate solution given by Timoshenko and Woinowsky–Krieger [37], except that different boundary conditions were used.

To determine the coefficients *α*′_v_ and *α*′_w_, the change in the cavity volume caused by the deflection of one of the component panes was analyzed, assuming that the rotational stiffness of the connection *C* [kNm/(m·rad)] is known. It was also assumed that the deflection of a pane with dimensions *a* × *b* (width × length) is the algebraic sum of the deflection of a simply supported pane loaded with a uniform load *q* and edge supporting moments (*M*_y_)_y=±b/2_ and (*M*_x_)_x=±a/2_ (Figure 3); see Equations (3)–(6) in [36] based on [37]. The unknown supporting moments were given as follows:(6)$${\left(M_{y}\right)}_{y=\pm b/2}=\sum_{i=1,3,5\ldots }(-1{)}^{(i-1)/2}\cdot F_{i}\cdot \cos\frac{i\pi x}{a}$$
(7)$${\left(M_{x}\right)}_{x=\pm a/2}=\sum_{i=1,3,5\ldots }(-1{)}^{(i-1)/2}\cdot G_{i}\cdot \cos\frac{i\pi y}{b}$$

*F*_i_ [kNm/m] and *G*_i_ [kNm/m] are strongly decreasing numerical sequences whose elements should be determined from the boundary conditions. In this case,
(8)$$\frac{\partial w}{\partial y}=\frac{M_{y}}{C}\;for\;y=\pm b/2\;\;\;\;\;\;\;\;\frac{\partial w}{\partial x}=\frac{M_{x}}{C}\;for\;x=\pm a/2$$

The analysis [36] results in the construction and solution of an appropriate system of linear equations in which the first five elements of both sequences are unknown. As a solution, it is convenient to provide dimensionless values of *F*′_i_ and *G*′_i_, such that
(9)$$F_{i}={F^{\prime }}_{i}\cdot q\cdot a^{2}\;\;\;\;\;\;\;G_{i}={G^{\prime }}_{i}\cdot q\cdot a^{2}$$

For practical reasons, another dimensionless quantity was also defined:(10)$$R_{c}=\frac{2\pi \cdot D}{a\cdot C}$$

Thanks to the use of this parameter, *F*′_i_ and *G*′_i_ depend only on the *R*_c_ value and the IGU aspect ratio *ε* = *b*/*a*. For the boundary values, *C* = 0, *R*_c_ **→** ∞ means simple support, and *C* **→** ∞, *R*_c_ = 0 means clamped fixity.

The *F*′_i_ and *G*′_i_ coefficients for selected *R*_c_ values and aspect ratios *ε* = *b*/*a* are presented in Appendix A. Knowing these values makes it possible to determine the coefficients *α*’_w_ according to Equations (3)–(6) from [36] and then, after integrating these equations, calculate the coefficients *α*’_v_. Tables with calculated dimensionless coefficients are presented in Appendix A.

It should be noted that for the boundary values (*R*_c_ **→** ∞, *R*_c_ = 0), the coefficients *F*′ and *G*′ (see Appendix A) describing the supporting moments and the coefficients *α*′_w_ regarding the maximum deflection are consistent (differences do not exceed 0.5%) with tables presented by Timoshenko and Woinowsky–Krieger in [37].

### 2.3. Stress in Elastically Supported Glass Panes

The stress at the center of the glass pane was estimated using well-known equations from the theory of plates. In general,
(11)$$\sigma_{x}=\frac{6\cdot m_{x}}{d^{2}}\;\;\;\;\;\sigma_{y}=\frac{6\cdot m_{y}}{d^{2}}$$
where
(12)$$m_{x}=-D\cdot \left(\frac{\partial^{2}w}{\partial x^{2}}+\mu \frac{\partial^{2}w}{\partial y^{2}}\right)\;\;\;\;\;m_{y}=-D\cdot \left(\frac{\partial^{2}w}{\partial y^{2}}+\mu \frac{\partial^{2}w}{\partial x^{2}}\right)$$

In this case, we use the principle of superposition:(13)$$w=w_{q}+w_{My}+w_{Mx}$$
where
*w*_q_, *w*_My_, *w*_Mx_—deflection of a simply supported pane under load *q* and deflections due to supporting moments *M*_y_, and *M*_x_ [kNm/m].

After integrating the appropriate functions of deflection (see Equations (4)–(6) in [36], based on [37]), for the center of the plate, we obtained
(14)$${\left(\frac{\partial^{2}w_{q}}{\partial x^{2}}\right)}_{x=0,\;\;y=0}=-\frac{4\cdot q{\cdot a}^{2}}{D\cdot \pi^{3}}\sum_{i=1,3,5,\ldots }\frac{{\left(-1\right)}^{\left(i-1\right)/2}}{i^{3}}\left(1-\frac{2+\beta_{i}\cdot th\beta_{i}}{2\cdot ch\beta_{i}}\right)$$
(15)$${\left(\frac{\partial^{2}w_{q}}{\partial y^{2}}\right)}_{x=0,y=0}=-\frac{2\cdot q{\cdot a}^{2}}{D\cdot \pi^{3}}\sum_{i=1,3,5,\ldots }\frac{{\left(-1\right)}^{\left(i-1\right)/2}}{i^{3}}\frac{\beta_{i}\cdot th\beta_{i}}{ch\beta_{i}}$$
(16)$${\left(\frac{\partial^{2}w_{My}}{\partial x^{2}}\right)}_{x=0,y=0}=-\frac{1}{D}\sum_{i=1,3,5\ldots .}{\left(-1\right)}^{(i-1)/2}\cdot F_{i}\cdot \frac{\beta_{i}\cdot \mathrm{th}\beta_{i}}{2\cdot \mathrm{ch}\beta_{i}}$$
(17)$${\left(\frac{\partial^{2}w_{My}}{\partial y^{2}}\right)}_{x=0,y=0}=-\frac{1}{D}\sum_{i=1,3,5\ldots .}{\left(-1\right)}^{(i-1)/2}\cdot F_{i}\cdot \frac{2-\beta_{i}\cdot \mathrm{th}\beta_{i}}{2\cdot \mathrm{ch}\beta_{i}}$$
(18)$${\left(\frac{\partial^{2}w_{Mx}}{\partial x^{2}}\right)}_{x=0,y=0}=-\frac{1}{D}\sum_{i=1,3,5\ldots .}{\left(-1\right)}^{(i-1)/2}\cdot G_{i}\cdot \frac{2-\gamma_{i}\cdot \mathrm{th}\gamma_{i}}{2\cdot \mathrm{ch}\gamma_{i}}$$
(19)$${\left(\frac{\partial^{2}w_{Mx}}{\partial y^{2}}\right)}_{x=0,y=0}=-\frac{1}{D}\sum_{i=1,3,5\ldots .}{\left(-1\right)}^{(i-1)/2}\cdot G_{i}\cdot \frac{\gamma_{i}\cdot \mathrm{th}\gamma_{i}}{2\cdot \mathrm{ch}\gamma_{i}}$$

In Equations (14)–(19), auxiliary quantities $\beta_{i}=\frac{i\cdot \pi \cdot \varepsilon }{2}$ and $\gamma_{i}=\frac{i\cdot \pi }{2\cdot \varepsilon }$ were used.

After inserting the appropriate values of partial derivatives into Equation (12) and then into Equation (11), it can be seen that the stress values in the pane center *σ*_x,cen_ [MPa], *σ*_y,cen_ [MPa] can be expressed using dimensionless coefficients *k*_x,cen_ i *k*_y,cen_. Then,
(20)$$\sigma_{x,cen}=k_{x,cen}\cdot \frac{6\cdot q\cdot a^{2}}{d^{2}}/1000\;\;\;\;\;\;\;\sigma_{y,cen}=k_{y,cen}\cdot \frac{6\cdot q\cdot a^{2}}{d^{2}}/1000$$

Dimensionless coefficients for calculating stress for selected *R*_c_ and *ε* = *b*/*a* values are presented in Appendix A. Calculations were made for two variants of the Poisson’s ratio: *μ* = 0.2 (according to [38]) and *μ* = 0.23 (according to [17]).

### 2.4. Reference IGU Parameters

For the numerical validation of the presented analytical model (Section 2.5) and the analysis of the influence of selected factors on static quantities (Section 3), a reference IGU with the following parameters was adopted: double-glazed IGU, dimensions *a* × *b* = 60 × 90 cm^2^ (*ε* = 1.5), thickness of glass panes *d*_ex_ = *d*_in_ = 4 mm, cavity thickness *s* = 16 mm. Initial conditions (state without deformations and stresses): *p*_0_ = 100 kPa, *T*_0_ = 293.15 K (20 °C). Glass parameters: *E* = 70 GPa, *μ* = 0.2.

### 2.5. Numerical Validation of the Analytical Model

In order to validate the computational model, the results of analytical calculations of a single glass plate (ANA) were compared with numerical calculations (NUM). The following static values were analyzed: maximum deflection *w*_max_, the stress in the plate center *σ*_x,cen_; *σ*_y,cen_ and edge stress *σ*_x,ed_; *σ*_y,ed_.

The numerical model was designed in Autodesk Robot Structural^TM^ using the “plate” structure type (Figure 4). The computational model was defined as a “shell”. Geometric dimensions and material data were determined according to Section 2.4. Linear supports were assumed on all edges. The support conditions are presented in Table 1. The plate was uniformly loaded with *q* = 5 kPa in the *z* direction. Due to the assumed vertical position of the plate, the self-weight load was omitted. The finite element mesh of size 1 × 1 cm^2^ was created using the simple (Coons) four-node method.

The calculation results are summarized in Table 2 and Table 3. Sample maps of static quantities are presented in Appendix A.

Based on the data presented in Table 1 and Table 2, it was found that for the key parameters describing the deflection and stress in the analyzed plate, the analytical results are in good agreement with the numerical ones, which proves the mathematical correctness of the proposed model. A similar agreement was obtained when analyzing other plate thicknesses and dimensions. Significant discrepancies were found only for edge stresses in the case of low *C*-values (marked gray). This is related to the failure to include in the analytical model the stresses from the reactions preventing the displacement of the plate corners during bending [37]. These additional stresses have a noticeable effect in the case of *R*_c_ > 2. Due to these discrepancies, coefficients for calculating edge stress are not given in Appendix A for *R*_c_ > 2 kNm/(m·rad), and results for *C* < 2 kNm/(m·rad) are not given in the graphs in the “Results” section.

However, the adopted simplification does not affect the further course of the analysis. In cases of close to simple support, the maximum stress (in terms of absolute value) occurs in the plate center and is much greater than at the edge. With the increase in rotational stiffness, the influence of edge stresses is increasingly stronger. In the analyzed plate, for *C* ≥ 5 kNm/(m·rad), the stresses *σ*_x_ are greater at the edges than in the center of the plate.

## 3. Results

### 3.1. Scope of Analysis

The analysis of the influence of rotational stiffness on static quantities in IGUs was performed in such a way that, based on the reference unit described in Section 2.4, one of the following parameters was changed: width of the glass pane *a* (0.4–1.2 m), aspect ratio *ε* (1.0–2.0), glass thickness *d* (4–10 mm), and cavity thickness *s* (12–18 mm). Double-glazed IGUs and two types of loading were assumed: symmetrical (change in atmospheric pressure) and asymmetrical (wind pressure). The static quantities analyzed included load reduction *r* [%], maximum deflection (in the center of the IGU) *w*_max_ [mm], and maximum stress (in the IGU center *σ*_x,cen_ [MPa], at the edges *σ*_x,ed_ [MPa]). Stresses *σ*_y_ were not analyzed because they are always smaller than *σ*_x_ (or equal to *ε* = 1.0). The resulting graphs illustrate the absolute values of the analyzed quantities. To demonstrate the increase in the resultant load with increasing *C*, the load reduction values are arranged on the ordinate in reverse order.

Based on the results of the tests presented in [10] (described in more detail in Section 4), the range of variation *C* = 0–5 kNm/(m·rad) was assumed. It was therefore considered that higher *C*-values are unlikely for real IGUs. To simplify the description of the results, the unit with simply supported panes was designated IGU(C0), while for *C* = 5 kNm/(m·rad), the symbol IGU(C5) was used.

### 3.2. Symmetrical Load—Atmospheric Pressure Changes

The load of atmospheric pressure drop of ∆*p* = 5 kPa was assumed, i.e., from *p*_0_ = 100 kPa to *p*_a_ = 95 kPa. Such pressure fluctuations are possible due to natural changes in weather conditions. It should be noted that similar effects would be caused by an increase in gas temperature in the cavity (for the reference IGU, the equivalent temperature increase is 15.4 K).

The reduction of the symmetrical load *r*_sym_ was calculated using the following formula:(21)$$r_{sym}=\left(1-\frac{q}{∆p}\right)\cdot 100\%$$
where *q* [kPa] is the resultant load, identical for both glass panes.

This reduction is related to the interaction of the gas enclosed in the cavity. It can be seen that as the sizes of IGUs increase, *r*_sym_ tends toward 100%. The calculation results are shown in Figure 5, Figure 6, Figure 7 and Figure 8. Spreadsheet printouts for all analyzed cases are presented in Appendix A.

As mentioned, the elastic support of the component panes at the edges leads to feedback. With the increase in rotational stiffness, the susceptibility of the panes to deflection decreases, which limits the reduction of the external load, i.e., the resultant load is higher. This is apparent in Figure 5a, Figure 6a, Figure 7a and Figure 8a. This phenomenon can affect the resulting values of the maximum deflection and stress in the IGUs in different ways.

Figure 5b,c indicates that increasing rotational stiffness is disadvantageous for large dimensions of IGUs. For example, for *a* = 120 cm, the deflection in IGU(C5) is 16.6% greater than in IGU(C0), and the stress in the glass pane center is even 58.7% greater. Additionally, edge stress begins to dominate at *C* = 2 kNm/(m rad). Fortunately, in large dimensions of IGUs, the stress values are not significant. As is generally known, the greatest stresses occur near the so-called characteristic (critical) widths. For the analyzed IGU, the critical width calculated on the basis of [8] is 37.2 cm. It can be seen that near this value (calculations for a = 40 cm), the stress in the pane center for IGU(C5) is only 2.0% greater than in IGU(C0), and the deflection is 15.0% smaller.

The analysis of different aspect ratios showed that the most advantageous variant for deflection is the square IGU (Figure 6b). The deflection in IGU(C5) is 2.2% smaller than in IGU(C0). The change of *ε*-value has no significant influence on the resulting dependencies concerning stress (Figure 6c). Edge stresses dominate for *C* values greater than 4 kNm/(m rad).

Figure 7b,c demonstrates that increasing rotational stiffness is beneficial for thicker glass panes. For example, for *d* = 10 mm, the deflection in IGU(C5) is 5.5% smaller than in IGU(C0), and the stress in the pane center is 2.9% smaller. The edge stress is significantly smaller. It can also be seen that the stress values are the highest for glass panes with a thickness of 6–8 mm.

Finally, the analysis of different gas cavity thicknesses (Figure 8b,c) details that this parameter does not have a significant effect on the percentage change in static values with increasing *C*. This is due to the fact that the impact of the cavity thickness on the deflection and stress in the IGUs is approximately linear. So, with increasing thickness, the deflection and stress increase, but the shape of the resulting graphs is similar.

### 3.3. Asymmetrical Load—Wind Pressure

A wind pressure load of 0.3 kN/m^2^, acting on the glass pane “ex”, was assumed, which corresponds to wind speed of about 80 km/h [39]. The reduction of this load *r*_win_ was calculated using the following formula:(22)$$r_{win}=\left(1-\frac{q_{ex}}{q_{z,ex}}\right)\cdot 100\%$$
where *q*_ex_ [kPa] is the resultant load acting on the glass pane “ex”.

This reduction is related to the transmission of part of the load from the glass pane “ex” to the glass pane “in” via the gas cavity. It can be seen that when increasing the IGU dimensions, the reduction of this load tends toward 50%, i.e., the external load is distributed on both panes in half. It should be added that in the case of different thicknesses of the component glass panes, the loads are distributed differently—proportionally to the stiffness of the component glass panes.

The analysis results are presented in Figure 9, Figure 10, Figure 11 and Figure 12. Spreadsheet printouts for all analyzed cases are presented in Appendix A.

First, it can be observed that for the load acting only on the “ex” glass pane, the deflection and stress in the pane center increase with increasing the IGU dimensions or decreasing the thickness of the component panes. Therefore, there is no critical width here. Secondly, the load reduction (Figure 9a, Figure 10a, Figure 11a and Figure 12a) is approximately half of that in symmetric loads (Figure 5a, Figure 6a, Figure 7a and Figure 8a). As a result, the increase in the resultant load caused by increasing the rotational stiffness is not as significant. Consequently, the increase in the *C*-value in each case considered leads to decreased deflection and stress in the pane center.

For different dimensions of IGUs (Figure 9b,c), within the range of the analyzed data, the deflection in IGU(C5) is 38.9 –57.6% smaller than in IGU(C0), and the stress in the pane center drops by 26.7–43.4%. However, it should be noted that at certain *C*-values, the edge stress starts to dominate (e.g., for *a* = 10 cm, this limit value of *C* is about 2.5 kNm/(m·rad).

For different aspect ratios (Figure 10b,c), the deflection in IGU(C5) is 42.1–53.4% smaller than in IGU(C0), and the stress in the pane center drops by 29.5–40.7%. For different pane thicknesses (Figure 11b,c), these values are 8.6–48.4% for deflection and 6.1–35.1% for stress. The change in the gas cavity thickness (Figure 12b,c) has almost no effect on the static values in IGU. The deflection drops by about 48–49%, and the stress in the pane center by about 35–36%. Edge stresses start to dominate at a *C* greater than about 4 kNm/(m rad).

## 4. Discussion

Estimating static values in IGUs is a complex issue. The efficiency of the interaction of the gas enclosed in the cavity depends on many factors related to the unit’s design and the assumed load combination. The proposed model introduces an additional factor of influence, namely the possibility of connecting the component panes with a spacer other than a simple support. It was assumed that the parameter describing this connection is the rotational stiffness C. Then, the methodology was developed to determine several dimensionless coefficients and apply them to the basic model describing IGUs with simply supported glass panes. The advantage of the model is the possibility of performing calculations using generally available spreadsheets, e.g., Microsoft Excel^TM^ (2016). Due to the mathematical complexity of the equations, setting up and testing such a spreadsheet requires quite some effort; however, after completing this task, the results are obtained automatically. A certain disadvantage of the model is the limitation of its application to the range of linear–elastic deflections of the glass pane.

This article analyzed only the most straightforward cases of IGU structures and acting loads to show general trends. However, it is not difficult to analyze any combinations of loads (wind, climatic loads) for multi-glazed IGUs with different stiffness of the component panes. It is also possible to declare different rotational stiffness for the individual panes of the unit.

Taking into account the conducted analysis in the literature, it can be noticed that the issue of rotational stiffness of the connection of component panes with the edge spacer in real IGUs is poorly understood; in particular, there are few experimental studies in this area. Based on original research [20] and earlier studies [40], Buddenberg et al. estimated that the rotational stiffness of the connection should not be greater than 0.2 kNm/(m·rad)—this is a state close to simple support. Also, Kozłowski et al. [32] show agreement of with the measured values of forced overpressure in the cavity with the values calculated for simple support. It can be noted, however, that these tests were conducted using IGU samples that were not embedded in the window frame.

As mentioned, the deflection tests in the IGU embedded in the frame were carried out by Stratiy [10]. Unfortunately, the author did not provide exact numerical values, but presented the results as a graph of the dependence of deflection on the value of forced load (Figure 6 in [10]). In the linear–elastic range, the measured deflection takes intermediate values between the estimated ones for simple support and restraint. Based on approximate values read from that graph, taking into account the parameters given in the article (dimensions 100 × 100 cm^2^, glass thickness 4 mm), the rotational stiffness was estimated at about 3.5 kNm/(m·rad). For this reason, in Section 3, it was assumed that a slightly higher value is possible for real IGUs in operating conditions.

In the context of the analyzed issue, attention should be paid to the variety of edge spacer designs used [5]. The classic solution is an aluminum or steel profile with a closed cross-section filled with a molecular sieve. Spacers with improved thermal insulation made of plastic with aluminum foil have a similar shape [41]. In most of these solutions, double sealing is used, e.g., polyisobutylene at the contact between the glass pane and the spacer (primary sealant) and a polyurethane or silicone sealant as a filling at the IGU edges (secondary sealant). Composite spacers made of structural foam [42] or a PIB/silicone system [33] are a separate type. These connections have viscoelastic properties [43], i.e., their efficiency depends on factors such as temperature, load duration, number of load repetitions, etc. Also, the aging of sealants may affect their properties [44].

Another factor that can be assumed to affect the rotational stiffness of the connection is the way the IGU is embedded in the frame. Various solutions are used here: rubber and silicone gaskets and bonded glazing systems [45]. Manufacturing and assembly errors may also have an impact—incorrectly selected dimensions of glass fillings mean that they are too shallowly, too deeply, or too tightly embedded in the frame [46]. Therefore, the freedom of rotation of the component glass on the support may be limited by the action of the window frame elements.

Taking the above into account, the author sees a need for experimental studies on the possibility of stiffening the connection of IGU component panes with different spacer structures, also using IGU samples embedded in the window frame. As shown in Section 3, under certain conditions, over-stiffening the connection can lead to increased deflection and stress in the IGUs.

## 5. Conclusions and Further Work

The paper implements the task of taking into account the possibility of elastic connection of component glass panes with the edge spacer to determine deflection and stress in IGUs. The proposed model is based on the previous analytical solution, which assumes simple support. Based on the Kirchhoff linear–elastic plate theory, appropriate coefficients were determined, which, when used in the basic model, allow estimation of the change in the gas cavity volume, deflection, and stress in IGUs, assuming any rotational stiffness of the connection of component panes with the spacer.

For a single plate loaded uniformly per area, the model was numerically validated for the entire range of *C*-values (from simply support to clamped fixity). Therefore, it can be helpful for the analysis of plates made of other homogeneous materials, not only glass.

The presented calculation examples concern the influence of rotational stiffness on static quantities for simple cases of climatic loads of double-glazed IGUs: change of atmospheric pressure (symmetrical load) and wind pressure (asymmetrical load). It was found that a kind of feedback loop occurs here—the deflection and stress in the center of a single elastically supported glass pane decrease with increasing rotational stiffness. On the other hand, the reduced susceptibility to deflection of the component panes weakens the interaction of the gas in the cavity, which leads to an increase in the resultant load acting on the component panes. As a result, the influence of rotational stiffness on the resulting static quantities can be different.

This influence was mostly negative for symmetrical loads in the analyzed examples (for *C* = 0–5 kNm/(m rad)). In some cases, the deflection and stress in the center were over 50% greater than for free support. However, this negative influence was insignificant for dimensions close to critical. A clearly positive influence of the connection stiffening was identified for wind load. In some cases, the static values were reduced by approx. 40 ÷ 50%. More complex load combinations and IGU structural arrangements require individual analysis.

The possibility of an elastic connection of component panes with the edge spacer in IGUs is poorly tested experimentally. Only a few studies are described in the literature. In particular, there is a lack of experiments regarding the possibility of stiffening this connection in IGUs embedded in the window frame, which can be mentioned as a direction for further research.

Of course, the analytical model requires continuous improvement. In the near future, the scope of the model will be extended to cover large deflections. This is very important because, in such cases, the susceptibility of the glass component panes to deflection decreases, which can increase the resultant load. In addition, it is planned to analyze the influence of certain phenomena on the mechanical behavior of IGUs, which have not been previously taken into account in analytical models: firstly, the possibility of thermal deflection of the component panes caused by the temperature difference on both surfaces of a single glass pane; secondly, the behavior of diagonally and horizontally located IGUs under radiative cooling conditions. This involves a temperature drop on the diagonal and horizontal surfaces of building envelopes below the outside air temperature on cloudless nights.

## Figures and Tables

**Figure 1 materials-17-04669-f001:**
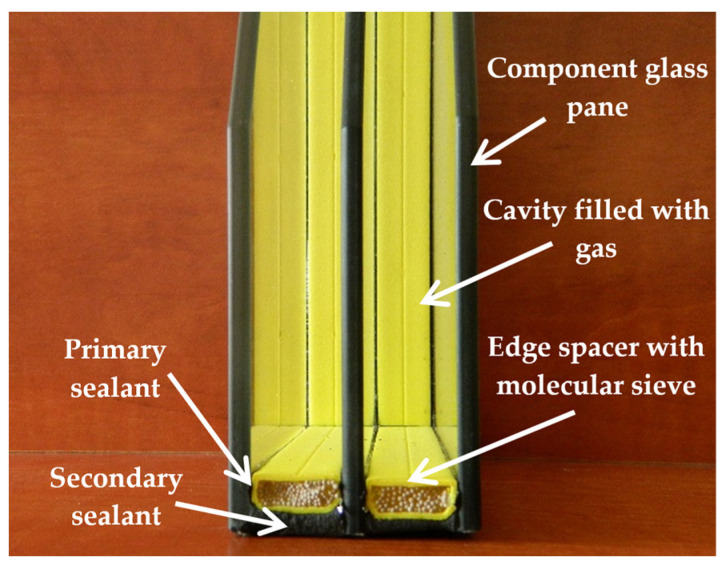
Structure of a typical IGU (author’s photo, IGU model from ©Swisspacer).

**Figure 2 materials-17-04669-f002:**
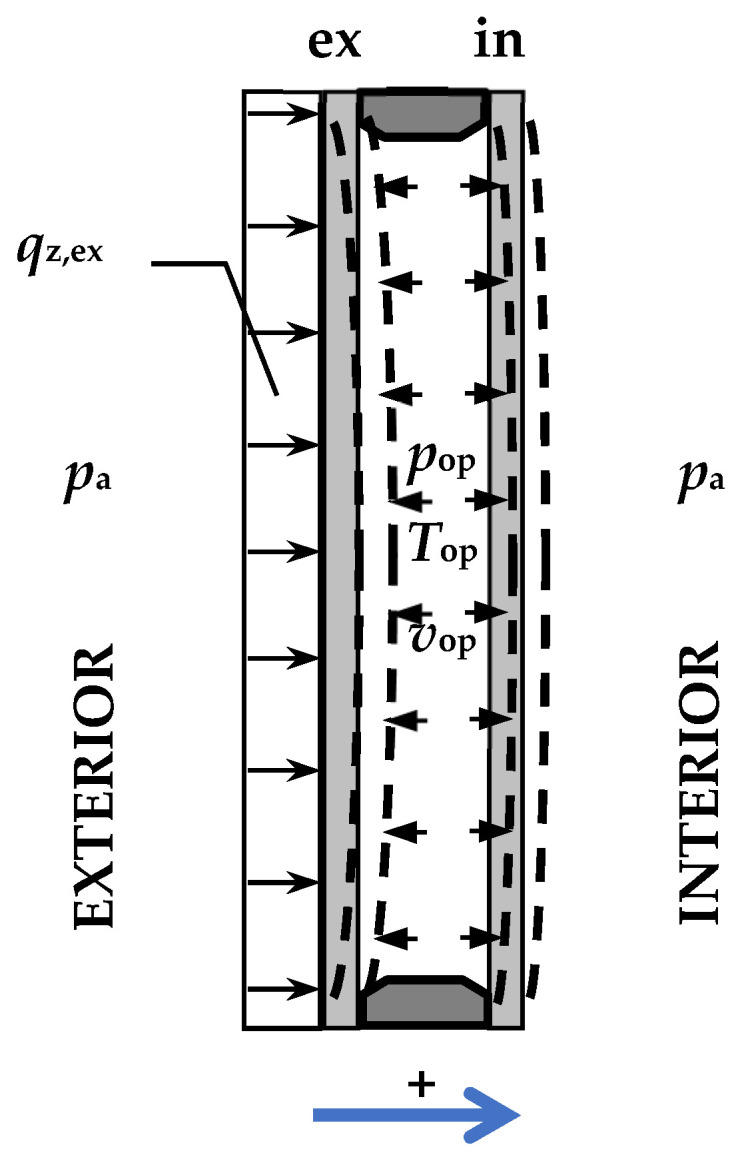
Double-glazed IGU—the symbols.

**Figure 3 materials-17-04669-f003:**
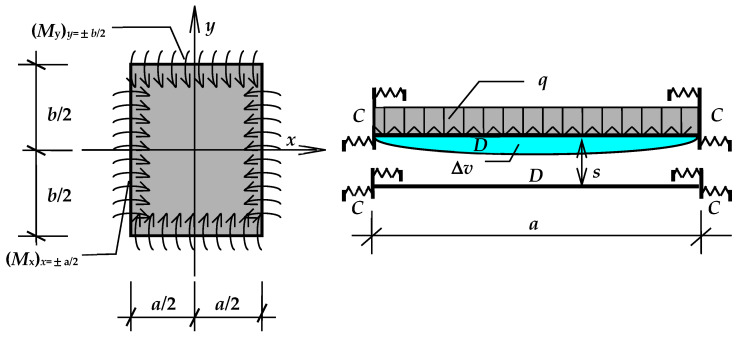
Diagram of a set of two glass panes elastically supported on the edges [36].

**Figure 4 materials-17-04669-f004:**
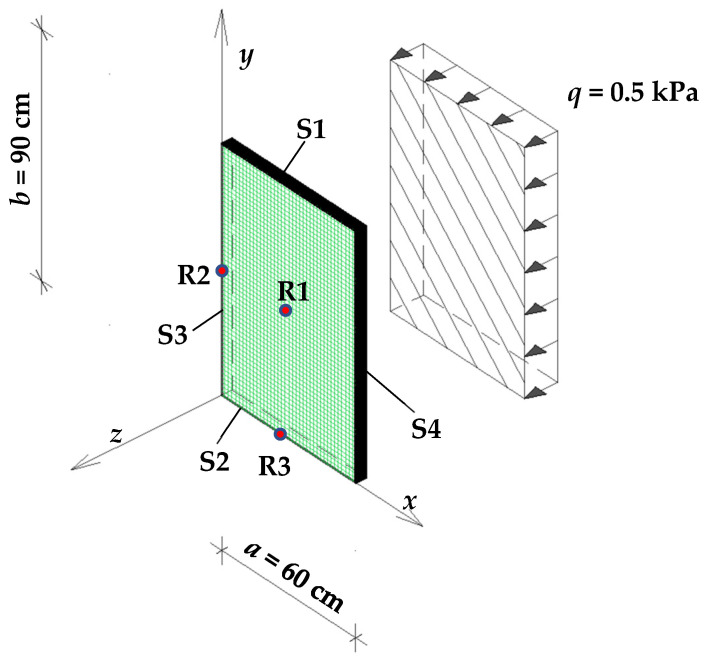
Diagram of the numerical model. S1–S4—support beams; R1—reference point for *w*_max_, *σ*_x,cen_, and *σ*_y,cen_; R2—reference point for *σ*_x,ed_; R3—reference point for *σ*_y,ed_.

**Figure 5 materials-17-04669-f005:**
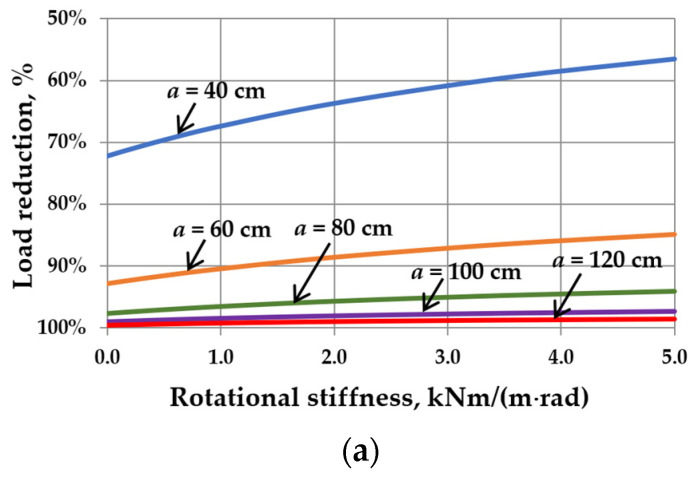
The influence of rotational stiffness on static quantities in IGUs loaded with a 5 kPa atmospheric pressure decrease for various widths of the unit: (**a**) load reduction; (**b**) maximum deflection; (**c**) maximum stress—the solid line marks *σ*_x,cen_, the dotted line marks *σ*_x,ed_.

**Figure 6 materials-17-04669-f006:**
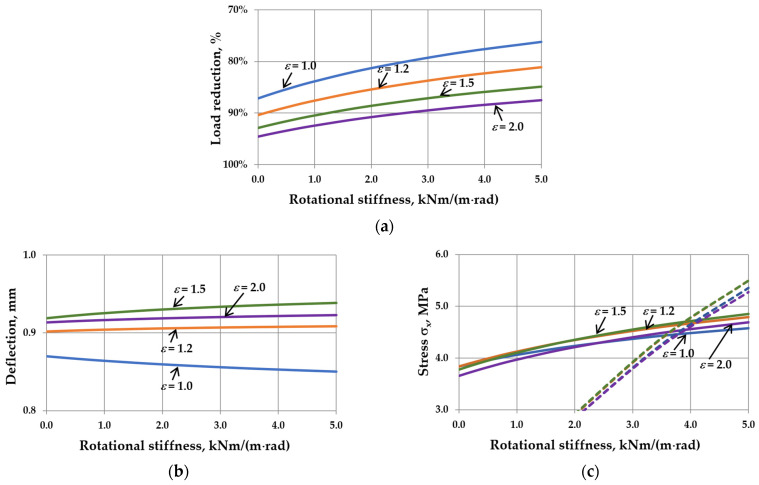
The influence of rotational stiffness on static quantities in IGUs loaded with a 5 kPa atmospheric pressure decrease for various aspect ratios of the unit: (**a**) load reduction; (**b**) maximum deflection; (**c**) maximum stress—the solid line marks *σ*_x,cen_, the dotted line marks *σ*_x,ed_.

**Figure 7 materials-17-04669-f007:**
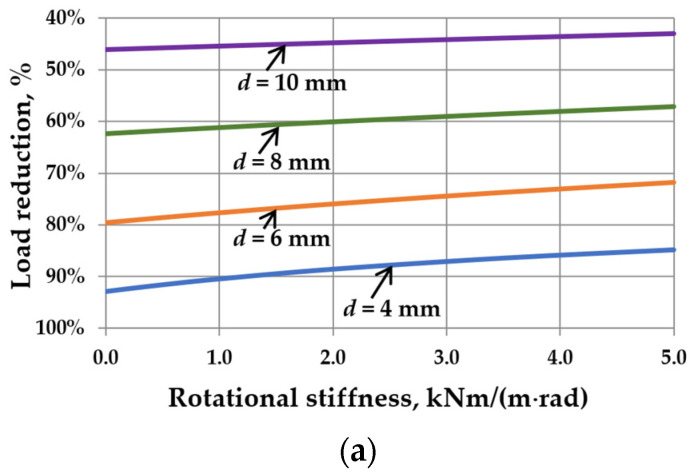
The influence of rotational stiffness on static quantities in IGUs loaded with a 5 kPa atmospheric pressure decrease for various thicknesses of glass panes: (**a**) load reduction; (**b**) maximum deflection; (**c**) maximum stress—the solid line marks *σ*_x,cen_, the dotted line marks *σ*_x,ed_.

**Figure 8 materials-17-04669-f008:**
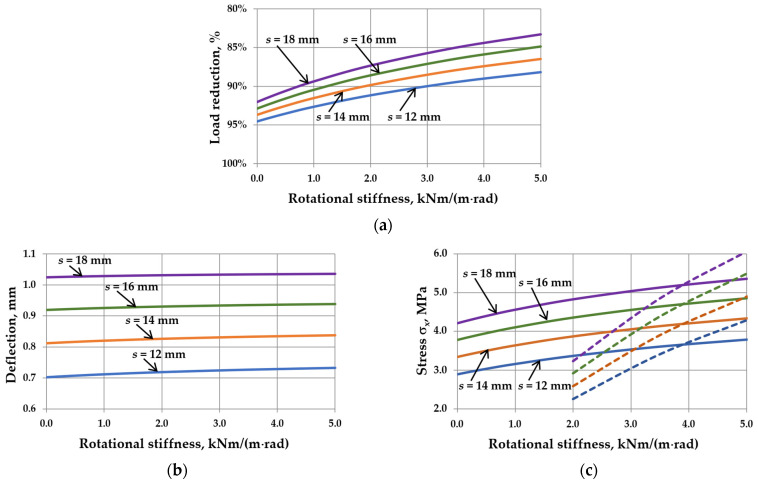
The influence of rotational stiffness on static quantities in IGUs loaded with a 5 kPa atmospheric pressure decrease for various thicknesses of gas cavity: (**a**) load reduction; (**b**) maximum deflection; (**c**) maximum stress—the solid line marks *σ*_x,cen_, the dotted line marks *σ*_x,ed_.

**Figure 9 materials-17-04669-f009:**
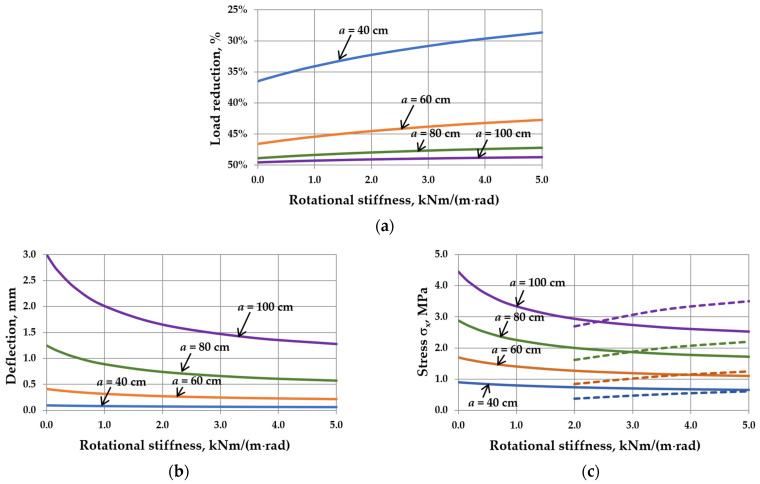
The influence of rotational stiffness on static quantities in IGUs loaded with a wind pressure of 0.3 kN/m^2^ for various widths of the unit: (**a**) load reduction; (**b**) maximum deflection; (**c**) maximum stress—the solid line marks *σ*_x,cen_, the dotted line marks *σ*_x,ed_.

**Figure 10 materials-17-04669-f010:**
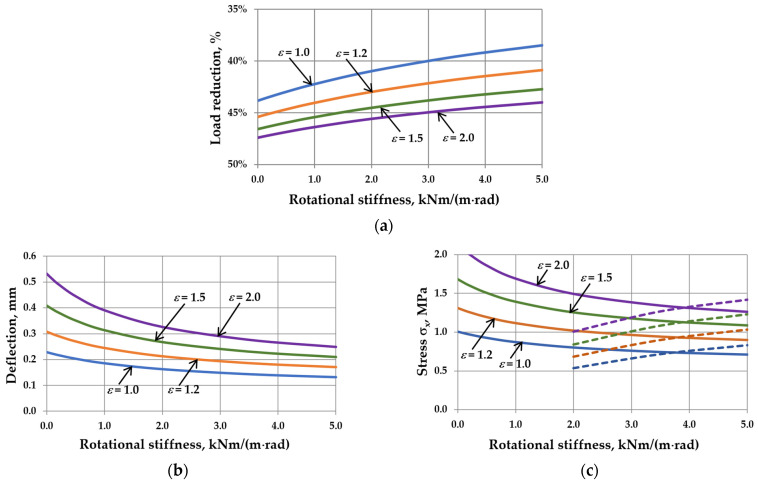
The influence of rotational stiffness on static quantities in IGUs loaded with a wind pressure of 0.3 kN/m^2^ for various aspect ratios of the unit: (**a**) load reduction; (**b**) maximum deflection; (**c**) maximum stress—the solid line marks *σ*_x,cen_, the dotted line marks *σ*_x,ed_.

**Figure 11 materials-17-04669-f011:**
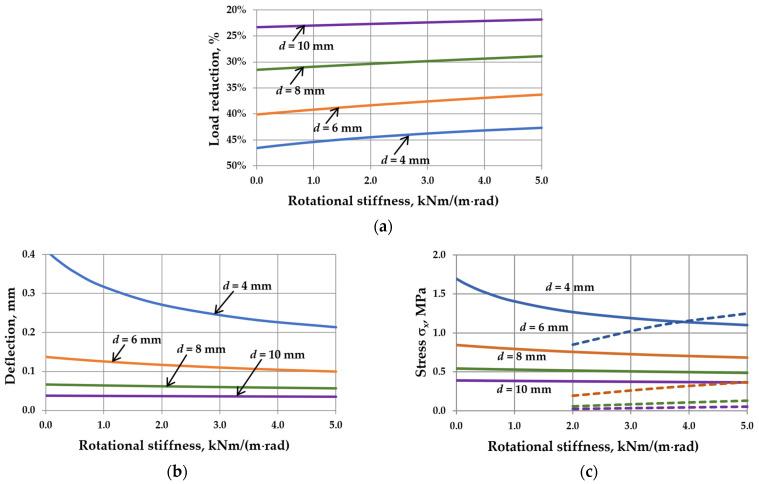
The influence of rotational stiffness on static quantities in IGUs loaded with a wind pressure of 0.3 kN/m^2^ for various thicknesses of glass panes: (**a**) load reduction; (**b**) maximum deflection; (**c**) maximum stress—the solid line marks *σ*_x,cen_, the dotted line marks *σ*_x,ed_.

**Figure 12 materials-17-04669-f012:**
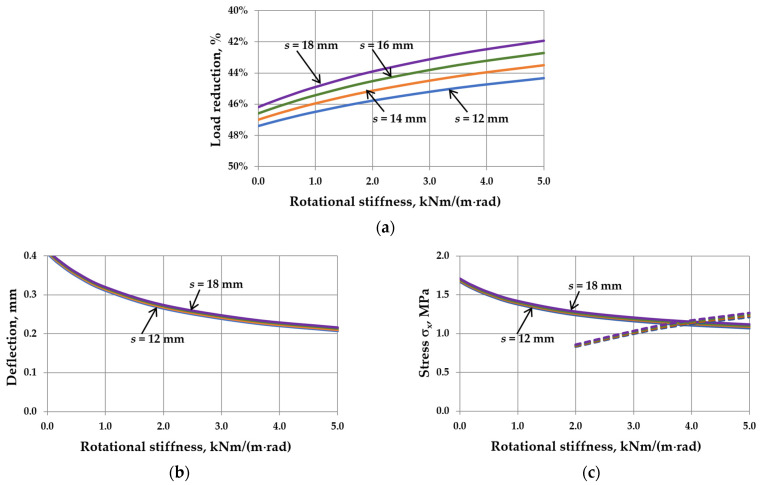
The influence of rotational stiffness on static quantities in IGUs loaded with a wind pressure of 0.3 kN/m^2^ for various thicknesses of gas cavity: (**a**) load reduction; (**b**) maximum deflection; (**c**) maximum stress—the solid line marks *σ*_x,cen_, the dotted line marks *σ*_x,ed_.

**Table 1 materials-17-04669-t001:** Support conditions.

Support	Rotational Degree of Freedom		Translational Degree of Freedom
	Direction *x*	Direction *y*	Direction *z*
S1 and S2	elastic, *C* (0, ∞)	fixed	fixed
S3 and S4	fixed	elastic, *C* (0, ∞)	fixed

**Table 2 materials-17-04669-t002:** Model validation. List of coefficients.

*C* [kNm/(m·rad)]	*R* _c_	*α*′_w_	*k* _x,cen_	*k* _y,cen_	*k* _x,ed_	*k* _y,ed_
0	**→**∞	0.007724	0.078420	0.042579	0.000000	0.000000
0.2	20.36217	0.007193	0.074308	0.040168	−0.006987	−0.005086
1	4.07243	0.005810	0.063601	0.033856	−0.025340	−0.018403
2	2.03622	0.004884	0.056447	0.029599	−0.037790	−0.027422
3.5	1.16355	0.004139	0.050711	0.026148	−0.047942	−0.034811
5	0.81449	0.003719	0.047477	0.024182	−0.053755	−0.039096
10	0.40724	0.003082	0.042590	0.021173	−0.062691	−0.045886
100	0.04072	0.002301	0.036614	0.017414	−0.073807	−0.055343
**→**∞	0	0.002197	0.035817	0.016907	−0.075267	−0.056765

**Table 3 materials-17-04669-t003:** Model validation. Comparison of analytical and numerical calculations.

*C* [kNm/(m·rad)]	*w*_max_ [mm]			*σ*_x,cen_ [MPa]			*σ*_y,cen_ [MPa]			*σ*_x,ed_ [MPa]			*σ*_y,ed_ [MPa]		
	ANA	NUM	diff.	ANA	NUM	diff.	ANA	NUM	diff.	ANA	NUM	diff.	ANA	NUM	diff.
0	1.287	1.287	0.00%	5.293	5.290	−0.06%	2.874	2.874	−0.01%	0.000	0.022	-	0.000	0.032	-
0.2	1.199	1.199	0.01%	5.016	5.013	−0.06%	2.711	2.711	−0.01%	−0.472	−0.493	4.60%	−0.343	−0.376	9.52%
1	0.968	0.968	0.01%	4.293	4.290	−0.06%	2.285	2.285	−0.01%	−1.710	−1.729	1.10%	−1.242	−1.269	2.19%
2	0.814	0.814	0.02%	3.810	3.808	−0.07%	1.998	1.998	−0.01%	−2.551	−2.569	0.69%	−1.851	−1.875	1.28%
3.5	0.690	0.690	0.02%	3.423	3.421	−0.07%	1.765	1.765	−0.01%	−3.236	−3.254	0.54%	−2.350	−2.371	0.89%
5	0.620	0.620	0.03%	3.205	3.202	−0.07%	1.632	1.632	0.00%	−3.628	−3.646	0.49%	−2.639	−2.659	0.75%
10	0.514	0.514	0.06%	2.875	2.873	−0.07%	1.429	1.429	0.01%	−4.232	−4.252	0.48%	−3.097	−3.116	0.59%
100	0.383	0.384	0.08%	2.471	2.470	−0.07%	1.175	1.178	0.24%	−4.982	−5.014	0.64%	−3.736	−3.757	0.58%
→∞	0.3660	0.3664	0.11%	2.418	2.416	−0.07%	1.141	1.145	0.32%	−5.081	−5.116	0.70%	−3.832	−3.857	0.65%

## Data Availability

Data are available upon request.

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
