# Peer review of "The Influence of Elastic Support of Component Glass Panes on Deflection and Stress in Insulating Glass Units—Analytical Model"

_materials, 2024, doi:10.3390/ma17184669_

Round 1

Reviewer 1 Report

Comments and Suggestions for Authors

The paper examines the impact of elastic edge support on the mechanical response of glass panes in insulating glass units (IGU). The elastic behaviour at the edges more accurately capture the influence of edge spacers on the behavior of glass panes in the IGU. The author introduces a parameter that characterizes the elastic support in relation to the rotational stiffness of the edge spacers. The article is well-organized, includes a comprehensive literature review, and clearly presents the results. The presented results on the impact of rotational stiffness of edge spacers influence in the mechanical analysis of glass panes provides new insights into the study of IGU.

I have the following minor remarks:

1)      Lines 39-41:The sentence  »For example, an increase in atmospheric pressure or a decrease in gas temperature results in a convex form of deflection of the IGU - the reverse action results in a concave form.«  needs to be revised. The words »convex« and »concave« need to be replaced.

2)      Line 273: “q=0,5 kN/m2” need to be “q=0.5 kPa” .

Reviewer 2 Report

Comments and Suggestions for Authors

The influence of elastic support of component glass panes on deflection and stress in insulating glass units is analysed in this paper. An analytical model is developed and presented. The work is interesting, however before it is published, I suggest some improvements.

In the numerical model, a scheme of the resolution process should be added. More details about the resolution process should also be added.

More information of the thermal process should be added. For example, some thermal phenomena as the solar radiation, convection, conduction and others that are used to evaluate the glass temperature. More information about the glass temperature evaluation should be added.

The cases studied and the input of each one should be added. Use as an example a table or a list.

The future evolutions of the work should be also added. In this point, some improvements to be analysed in future works can be added.

Comments on the Quality of English Language

Moderate editing of English language required.

Reviewer 3 Report

Comments and Suggestions for Authors

The paper is really good. Probably you can add some comments about the energetic implications and the possibility to take into account in the model of different types of emissivity of the internal glasses that influence the beahviour of the gap.

Comments on the Quality of English Language

Minor spelling errors was detected.

Round 2

Reviewer 2 Report

Comments and Suggestions for Authors

In the actual version, in general, all suggestions given by the reviewer were commented.